# Does green innovation moderate between FDI and environmental sustainability? Empirical evidence from South Asia

Awais Ahmed Brohi ◉ *◉, Yoshihisa Suzuki◉

Department of Economics, Graduate School of Humanities and Social Sciences, Hiroshima University, Hiroshima, Japan

◉ These authors contributed equally to this work.
* awais.brohi@yahoo.com

**Data Availability Statement:** Data is available at https://databank.worldbank.org/source/world-development-indicators.

## Abstract

The study aims to investigate how foreign direct investment (FDI) and green innovation (GI) impact environmental quality in South Asia. Moreover, this study examines the moderating role of GI between FDI and environmental sustainability. We use panel data from 1995 to 2018 for five South Asian nations namely, Pakistan, India, Bangladesh, Sri Lanka, and Nepal. For the empirical analysis, we used 1st generation cointegration like Pedroni and Kao, and 2nd generation cointegration tests like Westerlund. Moreover, for the long-run relationship, we employ fully modified least squares (FMOLS) and dynamic ordinary least squares (DOLS) estimation. The study's empirical results suggest that GI significantly enhances ecological sustainability in South Asian economies; however, FDI degrades the environmental quality. Furthermore, the results suggest that GI significantly moderates the nexus of FDI and ecological sustainability in South Asia. It is recommended that South Asian countries increase green innovation with FDI so that environmental quality can be assured for the region's sustainable development.

## 1 Introduction

Globalization has fostered foreign direct investment (FDI) in recent years; developing countries actively seek FDI inflows for economic development. The main objective of attracting more FDI is capital accumulation in the country [1]. The previous few decades have seen a rise in the manufacturing of highly advanced level products due to an increase in FD. FDI significantly affects host countries' technological advancements and infrastructure [2]. FDI has been hailed as the key economic growth tool, with researchers and practitioners seeing it as a reliable way for host economies to gain access to new technologies while also creating new jobs. The aggregate and sub-indices of infrastructure significantly promote the export and FDI inflow in the long run [3, 4]. FDI improves industry processes in host countries but may have negative effects on the natural environment. Although the host economies profit economically from production operations, the environmental cost may outweigh the financial gains in some cases. There are numerous types of research have been done on the link between FDI and

**Funding:** This study is funded by Professor Yoshihisa Suzuki who contributed in study design, data collection and analysis, decision to publish.

**Competing interests:** The authors have declared that no competing interests exist.

ecological sustainability; however, the results come to two conflicting conclusions. The first idea is the pollution haven hypothesis, which says that FDI increases environmental contamination. According to the pollution haven hypothesis put out by [5], ecologically damaging firms relocate from industrialized to developing nations due to stricter environmental restrictions at home. Whereas the second idea the "pollution halo hypothesis," contends that FDI enhances ecological health in developing nations through the latest technology and cleaner manufacturing.

As a result of their more fragile ecosystems, developing countries are increasingly concerned about ecological safety. This is because ecological protection by ensuring the safety of ecological resources has emerged as a major policy concern on a global scale. Sustainability in the environment has become a topic of heated discussion around the world. Both domestically and internationally, the countries are under intense pressure to improve their environmental protection measures. The developed world is more conscious of environmental issues. Therefore, these countries are adopting eco-friendly production methods in all sectors including manufacturing, agriculture, and services. Developing countries have less intention toward environmental protection strategies compared to developed countries. Keeping it in view, the countries are trying to gain FDI in green innovations. Investment in green technologies and green projects not only improves the economic benefits of the host country but also ensures environmental sustainability. Developing nations seek green technology innovation and eco-friendly production practices as a means of addressing environmental concerns. The concept of green innovation is popular in recent literature, which refers to environmentally friendly production and process [6, 7]. A clean atmosphere and reduced carbon dioxide emissions are two benefits that host economies can enjoy due to green innovation. It has been observed that the production technologies utilized by multinational corporations are more secure than the domestic technology utilized by emerging nations. These FDI also promote the energy infrastructure and increases the total factor of productivity [8–10].

South Asian region has attained incredible economic growth in current years, and these countries' contribution to global output is also noticeable. As per statistics from World Bank, the economic growth in South Asian countries has grown by 6% from 2015 to 2019. However, environmental pollution has also increased in these countries. Therefore, economic growth without ensuring the protection of the natural environment is unsustainable. Using fossil fuels is the major source of increasing ecological contamination in South Asia. The contribution of renewable resources in power creation is less in these countries due to insufficient advancement of technology [11]. Recently, South Asian countries are facing severe environmental damage. Besides, the other factors FDI is also proven a negatively affecting factor on ecological sustainability in these countries. Moreover, the level of green innovation at the firm level is not much satisfying in this region. Though, different studies have analyzed the nexus of FDI and GI on ecological sustainability in the case of different countries and regions. Such as the studies of [12–18].

The study aims to investigate how FDI affects the environmental quality in South Asia. Moreover, this study examines the moderating role of GI between FDI and environmental sustainability. This study also tries to investigate the research questions; does green innovation improve environmental quality? Does green innovation increase investment in the countries? The study contributes to the literature in several ways. First, no study has analyzed the moderation of green innovation in the association between FDI and ecological sustainability in the specific case of South Asia. Therefore, it is imperative to investigate the relationship between FDI and GI on ecological sustainability in the case of South Asian developing economies. Second, more specifically, this investigation complements the body of literature from two different angles direct and indirect effects. In direct effect, it explores the impact of FDI on

ecological sustainability. The second is an indirect effect which examines the FDI effect GI and GI affect ecological sustainability. Third, this study utilizes multiple econometric techniques like 1st generation and 2nd generation cointegration. First, we check the cross-sectional dependency, if it exists among the countries then we go for 2nd generation cointegration techniques. We also justify our results with other long-term cointegration techniques like FMOLS, and DOLS. Lastly, the study gives strong policy implications for the policymakers and investors for investment in the South Asian region.

The study has been organized in such a way that other parts of this study comprised a related literature review as a second part. The third part is about data and research methodology which includes the data collection, data presentation, and analysis techniques. The fourth part is about the discussion of empirical findings, whereas, the fifth part is about the conclusion which includes policy measures, limitations, and future areas of study.

## 2 Literature review

### 2.1 FDI and environmental quality

FDI has been extensively studied in the literature as a critical element of economic growth and development. A similar relationship has been investigated by several researchers, Such as studies of [19–24]. However, the link between FDI and ecological sustainability has received less attention. Several studies have found that FDI may impact the environment beneficially or negatively. One way that FDI might improve environmental performance is by introducing new technologies and management techniques. However, FDI can also result in increased resource depletion and pollution, especially in developing nations where laws may be laxer. The connection between FDI and air and water pollution has been the subject of numerous research. For instance, a study by [25] found that FDI in China reduced sulfur dioxide emissions but increased nitrogen oxide emissions negatively.

Shahbaz et al. [16] Studied the effect of FDI on ecological sustainability in countries with different levels of income countries. The findings of their investigation using the FMOLS model demonstrate the existence of the pollution haven hypothesis. Similarly, [26] looked into the impact of FDI and other variables on ecological sustainability in China in another study. Results from their study favored the pollution haven hypothesis that exists in China. In addition, [17] investigated the pollution haven hypothesis in 38 African countries from 1980 to 2014. Their empirical findings verify that FDI degrades the environment in these African economies, ensuring the presence of a pollution haven hypothesis. [27] investigate the environmental quality and health expenditure and found that positive environmental pollution shocks affect health expenditures positively in the long run, while negative environmental pollution shocks do not have a statistically significant effect on health expenditures. Positive and negative natural resource shocks affect health expenditures negatively in the long run.

FDI and ecological regulations have been the subject of other research. For instance, [28]. The study discovered that China's stricter environmental restrictions were linked to a decrease in the harmful impact of FDI on air contamination. Further research backs up the positive influence of FDI on ecological sustainability. They contend that FDI improves ecological health in the host nation by using contemporary, clean technologies and effective natural resource management. The pollution halo hypothesis states that FDI will decrease ecological degradation in the domestic country. [29] examined the connection between FDI and ES in OBOR nations. According to their study, FDI enhances environmental quality in these nations by lowering pollution thanks to developed R&D environments, superior technology, and trained human capital. Similarly, the results by [15, 30] also lend credence to the pollution halo theory. Additionally, [31] found that FDI had a detrimental influence on CO2 releases in

Algeria, despite their claim that FDI enhances Algeria's environmental quality and supports the pollution halo concept.

Additionally, several pieces of research discovered FDI's mixed or nonexistent influence on ecological sustainability. One of them, by Marques and Caetano [32], looked into how FDI affected environmental quality in various income-group countries. Their findings revealed that FDI improves environmental health in high-income countries while degrading the environment in medium-income ones. Opoku et al. [17] examined a similar link for 98 developing economies from various areas. Their research findings demonstrate a U-shaped link between FDI with CO2 discharges in developing Asian economies. However, FDI worsens ecological health in developing countries in Latin America, increasing its quality in Africa.

The literature generally states that the interconnection between FDI and ecological sustainability is complicated and can vary depending on the specific context. While FDI can bring in new technologies and management practices that can improve environmental performance, it can also lead to increased pollution and resource depletion. Stronger environmental regulations may alleviate the destructive effects of FDI on the ecology.

## 2.2 Green innovation and environmental quality nexus

Increasing environmental degradation has increased the need for green products and services. Therefore, green production through green innovations has increased. Ecological modernization theory states that the modernization of technologies can be helpful for the betterment of the natural environment. Henceforth, promoting green innovations will be helpful for climate mitigation. GI benefits environmental sustainability and can ensure sustainable development [33].

The simple definition of green innovation is the adoption of environmental techniques in the production process of the system. The core aim of green innovation is to protect the natural environment during the production of goods as well as services. The concept of green innovation is also used as eco-innovation or environmental innovation in literature [34]. Production of environmentally friendly products is known as green products, and green technology in production is called green process innovation. Green system innovation is related to environmentally friendly managerial and human capital stock. Technical innovation is usually gauged in patents and trademark applications. A patent is a unique measure of new technology secured by the country's law. Registering new patents gives a legal authority to produce that specific technique or technology. Similarly, GI is also gauged by the number of patents registered in environmentally friendly technology [35].

Most nations have recently been attempting to obtain more patents in green technologies. Green technology adoption is primarily driven by the need to combat rising environmental pollution. Numerous research, such as those by Carrión-Flores & Innes [36] and Jun et al. [35], have discovered that green innovation plays a favorable impact on enhancing environmental quality (2010). Eco-innovation can boost a country's economic performance and environmental health [37]. Researchers Cheng et al. [38] observed how GI affects ecological sustainability. According to their study, environmentally friendly innovation effectively uses natural resources to reduce environmental pollution and enhance environmental quality. In a similar vein, Villanthenkodath and Mahalik's [39] study discovered that technological innovation lowers environmental pollution in India. In a similar vein, Jin et al. [40] also examined how technological advancement will result in a decrease in environmental pollution.

Additionally, Song et al. [41] examined how environmental innovation reduces GHS. Their study's conclusions show that environmental innovations dramatically lower the GHS, enhancing the environment's quality. However, several other researchers, such as Shahbaz

et al. [26] and Demir et al. [27], have discovered that technological advancements increase environmental degradation by increasing CO2 emissions.

After careful consideration of the existing literature, we have found a literature gap that should be filled. We have not found any study in South Asia that investigated the moderating role of GI in connection with FDI and ecological quality. Therefore, this could fill up the literature gap on this issue. This study explicitly supports the pollution halo hypothesis through empirical evidence that green innovation has a significant moderating role between FDI and ecological sustainability.

## 3 Data and methodology

### 3.1 Data overview

The study investigates the moderating impact of green innovation between FDI and environmental quality. We assembled a panel of experts from five South Asian developing nations (Bangladesh, India, Pakistan, Sri Lanka, and Nepal) from 1995–2018. Due to data limitation of CO2 emissions which are accessible for 2018 so it is restricted to 2018. The metric tons of CO2 emitted per person is a proxy for EQ. Good ES is correlated with lower CO2 releases and vice versa. The total number of environmental patents applied for is used as an independent variable which is a proxy of GI. Furthermore, real GDP per capita is used to quantify economic growth, while FDI is measured using FDI inflows as a percentage of GDP. All this information was collected from the dependable source World development indicators (WDI), which is a publicly accessible source. The detailed data description is given in Table 1.

### 3.2 Methodology

**3.2.1 Model specification.** Considering the literature, we have identified that FDI, GI, and real GDP are the determinants of environmental quality. Therefore, these variables have been taken to examine the effect of FDI, GI, and real GDP on CO2 releases in the case of South Asia. The environmental quality in this study is gauged with CO2 emissions releases which state that more discharge of CO2 emissions means bad environmental quality. The base model is given below:

$$CO_{2i} = \int (FDI, , GI, RGDP) \tag{1}$$

Where CO2 emissions represent environmental quality, FDI is foreign direct investment, GI is green innovation, and RGDP is real GDP. The linear form of Eq 1 is given below.

$$CO_{2it} = \alpha_{it} + \beta_{it}FDI_{it} + \delta_{it}GI_{it} + \theta_{it}RGDP_{it} + \varepsilon_{it} \tag{2}$$

$\alpha_{it}$ and $\varepsilon_{it}$ are the regression slope and error term of the regression equation, respectively. The sign $i$ and $t$ correspond to years (1990–2018) and cross-sections (countries). Moreover, the coefficients are the concepts shown in $\beta_{it}$, $\gamma_{it}$, $\delta_{it}$, and $\theta_{it}$.

**Table 1. Variables description.**

| Variable | Symbol | Measurement unit | Source |
|---|---|---|---|
| Co2 emission | CO2 | Metric ton per capita | WDI |
| Green innovation | GI | Total number of patent applications | WDI |
| Foreign direct investment | FDI | % of GDP | WDI |
| Real GDP per capita | RGDP | GDP per capita (constant 2015 US$) | WDI |

**3.2.2 Econometric methodology.** The cointegration and long-term associations between the variables form the foundation of empirical analysis. Confirming the cross-section dependency between variables is crucial before evaluating the long-term relationship. Thus, we do a CD test on the data, and if CD is discovered, we will utilize the CIPS and CADF unit root tests from the second generation of panel unit tests. Additionally, we do the Pedroni, Kao, and Westerlund panel cointegration tests. Last, this work employs FMOLS and DOLS methods for a long-term estimate.

*3.2.2.1 Cross-sectional Dependence (CD).* Due to external variables, such as shock, unobserved components, and spatial reliance in one nation, the cross-sectional dependence problem in econometric research is discovered. It may have policy consequences in other countries [42]. Therefore, if a CD test is not conducted, the results may be biased and unpredictable [43]. This study uses the CD test from M. Pesaran [44] due to this problem.

*3.2.2.2 Unit root tests.* This study used Im-Pesaran-Shin (CIPS) M. H. Pesaran [45] and augmented Dickey-Fuller (CADF)" test to analyze the unit root in the data. These techniques are used to address the cross-section dependence problem. The null hypothesis of these techniques can be rejected if all variables are uniformly stationary.

*3.2.2.3 Cointegration analysis.* The Pedroni [46], Kao [47], and Westerlund [48] tests, among others, were all implemented as residual-based panel cointegration tests. The Westerlund [48] test does not need to be corrected for temporal dependence and is appropriate for residual panel heterogeneity and cross-sectional dependence. We looked at the panel cointegration form as an alternate hypothesis to test the cointegration possibilities.

$$\rho_{it} = \mu_i \rho_{it-1} + \hat{\gamma}_{it} \tag{3}$$

Panel variance ratio (PVR) and group mean-variance ratio (GVR) statistics are two variance ratio (VR) test statistics that Westerlund gives. They are both obtained from Eq 3. Both VR tests are based on the work of Breitung and M.H. Pesaran [49].

*3.2.2.4 The Slope of heterogeneity test.* To ascertain whether the association between an IV and a DV is similar across various subgroups of the data, a statistical test known as the slope of heterogeneity is used. The test is used to determine whether the IV and the DV relationship is the same for all subgroups if the slope coefficients in a regression model are identical across subgroups. If the equal slopes hypothesis is not accepted, distinct subgroups have different relationships between the IV and DV.

The slope of heterogeneity test can be performed by adding interaction terms to a regression model representing the product of the IV and a dummy variable indicating each observation's subgroup membership. The coefficients on these interaction terms can be used to test the hypothesis of equal slopes, and if the hypothesis is not approved, it shows the occurrence of slope heterogeneity. We have used the Pesaran Yamagata [50] test to verify the slope heterogeneity.

*3.2.2.5 Long-run estimators.* For the long-run estimation, we used fully modified-OLS (*FMOLS*) of Pedroni [46]. FMOLS equation is as follows:

$$\hat{\gamma}_{FMOLS} = \left[ \frac{1}{N} \sum_{i=1}^{N} \sum_{t=1}^{T} (J_{it} - \bar{J}_i)^2 \right]^{-1} \times \left[ \sum_{t=1}^{T} (J_{it} - \bar{J}_i)\hat{k}_{it} - T\hat{\partial}_{eu} \right] \tag{4}$$

*J, and k* are used as the DV and IV in this analysis, respectively. The Kernel estimator, which is used to determine the sequential connection of the covariance term, is denoted by the symbol $\hat{\partial}_{eu}$. In the equation, N and T stand for years and CS, respectively. The FMOLS approach can address heteroscedasticity by utilizing the Bartlett and Kernel processes [51].

The FMOLS $\hat{\partial}_{eu}$ term can also manage possible endogeneity and autocorrelation problems and deliver reliable findings [52]. However, it can estimate cointegrated panels but cannot solve the cross-sectional dependence issue [53]. We employ dynamic OLS (DOLS) as a second estimator in addition to FMOLS, as proposed by Stock and Watson [54]. Alternative order variables can be included with DOLS, which also handles the simultaneity of independent variables. Compared to other long-run estimators, DOLS is particularly helpful in small samples [54].

## 4 Estimation results

### 4.1 Descriptive statistics results

Extensive data can be summarized, arranged, and presented in a meaningful and understandable fashion using descriptive statistics. Calculating measures of central tendency, measures of variability, and frequency distributions are all part of descriptive statistics. The purpose of descriptive statistics is to provide a general overview of the data by describing the key characteristics of a dataset, such as patterns, trends, and correlations between variables. The findings of descriptive statistics are given in Table 2. The results show that the mean and standard deviation for CO2 discharges are -0.653 and 0.791, respectively. The mean and SD of GI are 2.294 and 0.575, and the mean and SD of FDI are -0.391 and 1.104. In addition, the mean and SD of RGDP are 7.604 and 1.851. Fig 1 show the graphical presentation of descriptive statistics in box plot and scatterplot diagrams.

### 4.2 Correlation matrix results

A table showing the correlation (CR) between various variables is named a correlation matrix (CRM). The correlation coefficient, which ranges in value from -1 to 1, is contained in each cell of the table and represents the strength and direction of the linear link between the two variables. If the CR between two variables is positive, this means that both variables increase or decrease at the same time, but if the CR between two variables is negative, the other variable rises while the first one falls. No linear relationship exists between the variables, as indicated by a CR coefficient of 0.

CRM is a valuable tool for summarizing the relationships between multiple variables and for identifying pairs of variables that are highly correlated. It is commonly used in finance, economics, and psychology to understand the relationships between multiple variables and make predictions based on this information.

The results of correlation coefficients exhibit that there is an inverse association between GI and CO2 emissions. Moreover, the FDI and RGDP are positively related to Co2 discharges see Table 3.

### 4.3 Cross-sectional dependence test results

The results of the CSD test are presented in Table 4. The null hypothesis has been rejected as per the results, indicating that CSD exists in the selected panel of South Asian developing economies.

**Table 2. Results of descriptive statistics.**

| Variable | Obs | Mean | Std. Dev. | Min | Max |
|---|---|---|---|---|---|
| Co2 emission (log) | 120 | -0.686 | 0.791 | -2.361 | 0.587 |
| Green innovation (log) | 120 | 2.294 | 0.575 | 0.029 | 3.167 |
| Foreign direct investment (log) | 120 | -0.391 | 1.104 | -5.298 | 1.299 |
| Real GDP per capita (log) | 120 | 7.604 | 1.851 | 5.990 | 13.008 |

Source: Author Calculated.

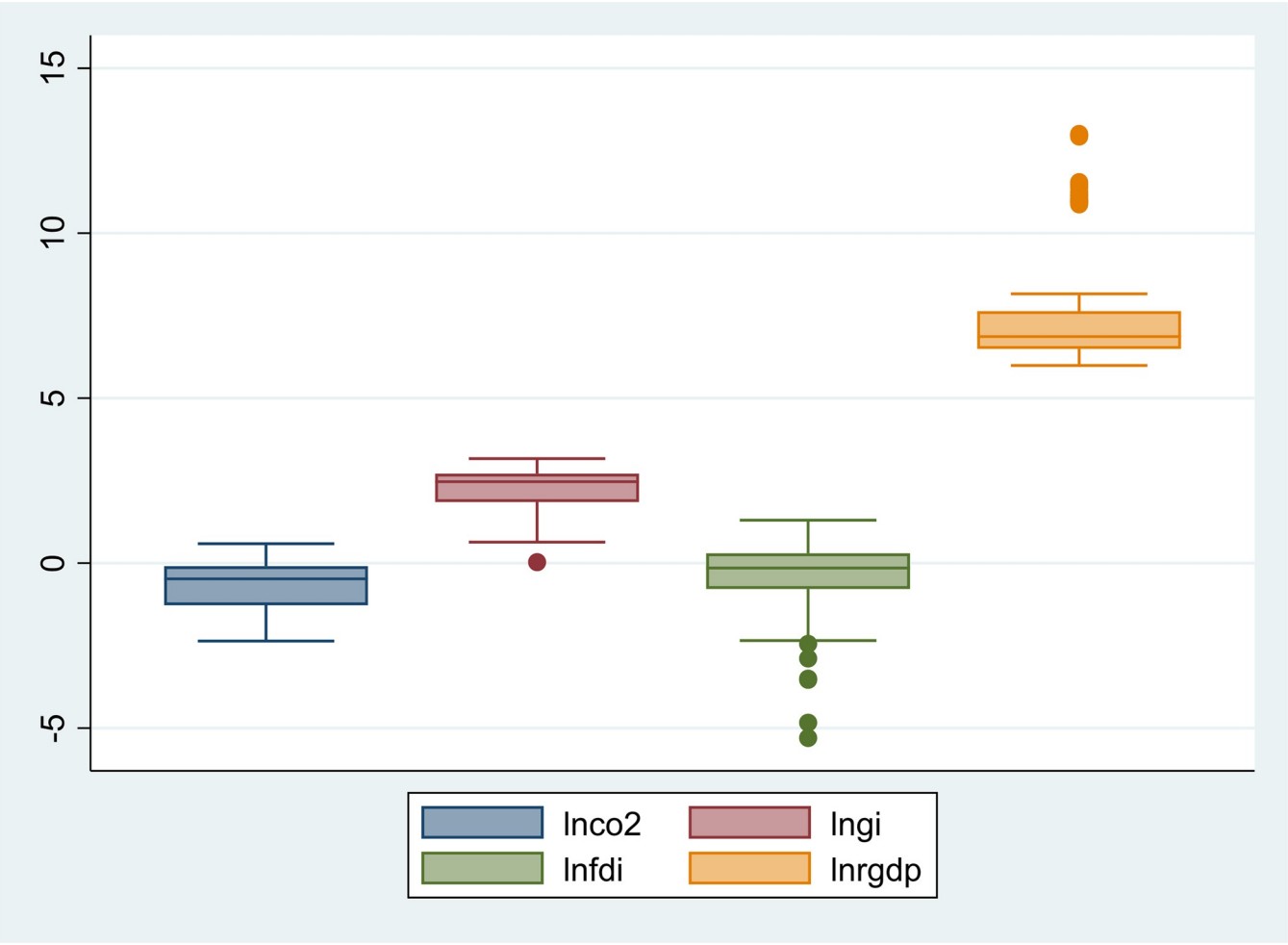

**Fig 1. Box plot of variables deleted separate file uploaded with submission generated from pace.**

## 4.4 Slope homogeneity test results

When variables are cross-sectional dependent then we check the homogeneity of the variables. The findings of SHT are given in Table 5. As per the results, the null hypothesis is not accepted. Therefore, the panels are confirmed as heterogonous. After the confirmation of heterogenous then we should go for the 2$^{nd}$ generation cointegration test.

## 4.5 Results of unit root tests

As CSD's existence is confirmed through the CSD test, and heterogeneity also exists then we used second-generation unit root tests, i.e., CIPS and CADF. The results of CIPS and CADF

**Table 3. Results of correlation matrix.**

| Variables | CO2 | GI | FDI | RGDP |
|---|---|---|---|---|
| CO2 Emissions | 1 | | | |
| Green patents | -0.001 | 1 | | |
| Foreign direct investment | 0.596 | -0.011 | 1 | |
| Real GDP per capita | 0.243 | 0.222 | 0.098 | 1 |

Source: Author Calculated.

**Table 4. Results of the cross-sectional dependence test.**

| Variable | CD-statistic |
|---|---|
| $CO_2$ | 12.88*** |
| GI | 13.42*** |
| FDI | 2.00** |
| RGDP | 15.48*** |

Notes: Under the null hypothesis of the independence of cross-sections of CD ~ N (0,1)

**,*** indicate significance levels at 5% and 1% respectively.

**Table 5. Results of slope homogeneity test.**

| | Δ Statistics | p-value |
|---|---|---|
| $\bar{\Delta}$ | 2.018 | 0.044** |
| $\bar{\Delta}_{adj}$ | 2.268 | 0.023** |

Note

** indicates significant at 5%. H0: slope coefficients are homogenous

tests are given in Table 6. As per the results, some variables are stationary at a level. However, all variables are stationary at first difference. Hence, it is established that we can use FMOLS and DOLS techniques to estimate the link between IV and DV in this study.

## 4.6 Results of panel cointegration tests

In this study, we used 1st and 2nd generation cointegration tests to check the long-run relationship. As CD test results confirm that there is cross-section dependence between variables so there must apply 2nd generation cointegration test. For results validation, we have used three cointegration tests, including KAO, and Pedroni for 1st generation and Westerlund for 2nd generation cointegration.

The results of the KAO test in Table 7 confirm that there is cointegration among the variables. The results revealed that there is a long-term relationship between FDI and environmental sustainability when green innovation mediates between them.

Similarly, the results of the Pedroni test also confirm the existence of cointegration among variables (see Table 8). This result is consistent with the Kao cointegration test. There is a long-run relationship between FDI and environmental sustainability.

Lastly, we applied Westerlund's (2005) cointegration technique to check the cointegration 2nd generation. Table 9 below shows the Westerlund cointegration which revealed that there is

**Table 6. Pesaran CADF & CIPS tests of unit roots.**

| Variable | CADF | | CIPS | |
|---|---|---|---|---|
| | Level | 1st-difference | Level | 1st-difference |
| $CO_2$ | -1.095 | -2.490[b] | -1.390 | -4.231[a] |
| GI | -2.828[a] | -4.319[a] | -3.311[a] | -5.026[a] |
| FDI | -3.084[a] | -3.957[a] | -2.894[b] | -4.392[a] |
| RGDP | -0.560 | -2.852[a] | -0.303 | -3.869[a] |

**Note**: significance at a = p<0.01, b = p<0.05, c = p<0.1

**Table 7. Results of KAO (1999) cointegration test.**

| | |
|---|---|
| Modified Dickey-Fuller t | -4.6618[a] |
| Dickey-Fuller t | -3.4966[a] |
| Augmented Dickey-Fuller t | -1.7803[b] |
| Unadjusted modified Dickey-Fuller t | -5.4278[a] |
| Unadjusted Dickey-Fuller t | -3.6619[a] |

**Note:** significance at a = p<0.01, b = p<0.05, c = p<0.1

a long-run relationship between FDI and environmental sustainability in the presence of green innovation as moderator.

## 4.7 Results of FMOLS estimation

We used the FMOLS estimation to estimate the long-run association of FDI, GI, and ecological sustainability. The results of FMOLS in Table 10 state that green innovation significantly reduces CO2 discharges with a coefficient value of -0.195, which confirms that with an increase in green patents in the case of South Asia the ecological quality will be improved. This finding is in line with the study of Jun et al. [35]. Increasing use of non-renewable energy has increased the environmental pollution in these countries; however, this is imperious for South Asian countries to increase green industrialization through green innovations at the firm level. This can be done through introducing environmentally friendly environmental policies such as carbon taxes, green credits, and green training to the employees of corporates. South Asian countries are developing in nature; therefore the shortage of capital investment in green initiatives is much more difficult for firms. Besides that, the risk of green investment is also a major concern for these countries. Therefore, these countries need international collaboration in green innovation investment projects, and also firms of these countries need domestic government support.

Furthermore, the results show that FDI degrades ecological quality in South Asia by increasing CO2 discharges with a coefficient value of 0.664, indicating that FDI reduces environmental quality. This finding is similar to the findings of Ali et al. [12] in the case of BRICS economies. FDI is deliberated as a major stimulator of the economy in South Asian countries like other developing countries. However, at the same time, FDI is also proven to negatively affect factors of the natural environment. Therefore, governments in South Asian countries

**Table 8. Results of the Pedroni cointegration test.**

| | |
|---|---|
| Modified variance ratio | -2.424[a] |
| Phillips-Perron t | -0.637 |
| Augmented Dickey-Fuller t | -1.791[b] |

**Note:** significance at a = p<0.01, b = p<0.05, c = p<0.1

**Table 9. Results of Westerlund (2005).**

| | Statistic | p-value |
|---|---|---|
| Variance ratio | 4.5614 | 0.0000 |

Null hypothesis = no cointegration

**Table 10. Results of FMOLS estimation.**

| Dependent variable = $CO_2$ emissions | | |
|---|---|---|
| Variable | Model 1 | Model 2 |
| GI (log) | -0.195[b] | -0.216[b] |
| FDI (Log) | 0.664[a] | 1.415[a] |
| RGDP (log) | 0.001 | 0.008 |
| FDI*GI | | -0.324[b] |
| $R^2$ | 0.480 | 0.492 |

**Note:** significance at a = $p<0.01$, b = $p<0.05$, c = $p<0.1$

should ensure ecological sustainability while seeking more FDI for long-term sustainable economic growth.

However, economic growth has a positive yet insignificant influence on CO2 discharges with a coefficient value of 0.001. Moreover, the results of the moderation analysis state that GI significantly moderates the association between FDI and CO2 discharges -0.324, which implies that FDI along with GI can better off the ecological quality in South Asia. Therefore, FDI in green initiatives in South Asian countries not only boosts economic growth but also improves ecological health.

### 4.8 Dumitrescu Hurlin panel causality tests

To check the multicollinearity among the variables we test the causality test. We use [55] causality test. Below Table 11 show the causality results which explain that there is no causality among the variables. Only real GDP cause CO2 and CO2 causes real GDP. These 2 variables have bi-directional causality with each other. The remaining variables have no such relationship. So we conclude that there is no multi-issue among the variables.

### 4.9 Robustness analysis through DOLS estimation

We conduct a DOLS test to verify the robustness of the results of the FMOLS estimation. The findings from DOLS are also in line with the findings of FMOLS (see Table 12). This exhibits

**Table 11. Pairwise Dumitrescu Hurlin panel causality tests.**

| Null Hypothesis: | W-Stat. | $\bar{Z}$-Stat. | Prob. | Outcome |
|---|---|---|---|---|
| LNFDI does not homogeneously cause LNCO2 | 1.414 | 0.445 | 0.656 | No Causality |
| LNCO2 does not homogeneously cause LNFDI | 1.278 | 0.261 | 0.794 | |
| LNGI does not homogeneously cause LNCO2 | 0.379 | -0.962 | 0.335 | No Causality |
| LNCO2 does not homogeneously cause LNGI | 1.562 | 0.647 | 0.517 | |
| LNRGDP does not homogeneously cause LNCO2 | 3.155 | 2.814 | 0.0049*** | Bi-Directional Causality |
| LNCO2 does not homogeneously cause LNRGDP | 3.374 | 3.112 | 0.0019*** | |
| LNGI does not homogeneously cause LNFDI | 1.230 | 0.195 | 0.8448 | No Causality |
| LNFDI does not homogeneously cause LNGI | 1.126 | 0.053 | 0.9575 | |
| LNRGDP does not homogeneously cause LNFDI | 0.389 | -0.949 | 0.3425 | No Causality |
| LNFDI does not homogeneously cause LNRGDP | 0.384 | -0.955 | 0.3395 | |
| LNRGDP does not homogeneously cause LNGI | 1.155 | 0.092 | 0.9262 | No Causality |
| LNGI does not homogeneously cause LNRGDP | 1.013 | -0.099 | 0.9204 | |

Source: Author Calculated. *** indicates significance at 1% = $p<0.01$.

**Table 12. Results of DOLS estimation.**

| Dependent variable = $CO_2$ emissions | | |
|---|---|---|
| **Variable** | **Model 1** | **Model 2** |
| GI (log) | -0.186[b] | -0.230[b] |
| FDI (Log) | 0.509[a] | 1.211[a] |
| RGDP (log) | -0.003 | 0.009 |
| FDI*GI | | -0.286[b] |
| $R^2$ | 0.456 | 0.502 |

**Note:** significance at a = $p<0.01$, b = $p<0.05$, c = $p<0.1$

that green innovation significantly reduces CO2 emissions with coefficient values of -0.186 and ensures to improve the ecological health in the South Asian region. Furthermore, results also show that FDI worsens the ecological quality in these economies by increasing CO2 emissions with coefficient values of 0.509. However, results further indicate that real GDP has an insignificant impact on ecological sustainability. Moreover, the results of the moderation analysis state that GI significantly moderates the association between FDI and ES in South Asian developing economies.

## 5 Conclusion and recommendations

Numerous researchers have looked into the association between FDI and environmental sustainability, but the findings have not been clear-cut. This study aims to understand better how foreign direct investment (FDI) affects environmental quality and how green innovation (GI) moderates South Asia's developing economies. We use data from 1995 to 2018 for five South Asian nations. For the empirical analysis, we used cross sessions dependence tests, second-generation unit tests, and cointegration tests, including Pedroni [46], Kao [47], and Westerlund [48]. Moreover, for the long-run relationship, we employ FMOLS and DOLS estimation. The study's empirical results suggest that GI significantly enhances ecological sustainability in South Asian economies; however, FDI degrades the environmental quality.

Furthermore, the results suggest that green innovation significantly moderates the nexus of FDI and ecological sustainability in South Asia. The main findings are in line with the study of Ali et al. [12]. According to their findings, GI is a factor that improves the EQ in a group of BRICS economies. However, the FDI degrades the ecological standards of these countries. Similarly, the study of Zamir et al. [14] concluded that using environmentally friendly technology, FDI, and renewable green usage increases economic growth in South Asian economies. However, environmental pollution has a detrimental effect on economic growth.

The study offers the policy implication for the policy maker and investors. As our study has added to the literature by analyzing the moderating effect of GI in the nexus of FDI and EQ as the developing countries of South Asia seek more FDI due to a shortage of capital and investment. However, environmental degradation has also been a significant concern in these countries due to increased FDI, industrialization, urbanization, and population growth. Therefore, it is recommended that South Asian countries make stricter environmental regulations for foreign investors and focus more on green innovations so that environmental quality can be assured for sustainable development in the region. To achieve better results for green growth, governments are advised to prioritize increasing investment in labor-related technical innovations, environmental technological innovations more particularly, technological breakthroughs connected to renewable energy and spending more on the R&D sector. From a policy perspective, the study argues that policymakers should not undervalue the benefits of

human capital on green growth since human capital helps to enhance green growth. A foundation for green growth is created by increased educational attainment since it increases ecological awareness and the effectiveness of the environment during the production and consumption processes. Additionally, the expansion of green practices in the tourist industry, such as green management, energy, and transportation, can help reinforce the positive effects of the sector on environmental quality in Asian countries. Government officials and policy-makers should revamp Asian ecotourism regulations. Thus, by reducing CO2 emissions, Asian nations might boost the tourist sector.

In this study, we have used green innovation as a moderating variable; it can be used as a mediating variable in future studies. Also, the impact of FDI on GI can be investigated by using environmental regulations as a moderating variable. Furthermore, this research is conducted on five developing economies of South Asia; however, this research could be extended to investigate the same issue in other developed and developing economies for a better understanding of the matter. Moreover, the data used in this study is until 2018, and more updated data can be used in future studies.

## Author Contributions

**Conceptualization:** Awais Ahmed Brohi.

**Data curation:** Yoshihisa Suzuki.

**Formal analysis:** Yoshihisa Suzuki.

**Funding acquisition:** Yoshihisa Suzuki.

**Methodology:** Awais Ahmed Brohi.

**Resources:** Yoshihisa Suzuki.

**Software:** Awais Ahmed Brohi.

**Supervision:** Yoshihisa Suzuki.

**Writing – original draft:** Awais Ahmed Brohi.

**Writing – review & editing:** Yoshihisa Suzuki.

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
