## [Decision Letter · Decision Letter 0]

21 Mar 2023

PONE-D-23-05630Does Green Innovation Moderate between FDI and Environmental Sustainability? Empirical Evidence from South AsiaPLOS ONE

Dear Dr. Brohi,

Thank you for submitting your manuscript to PLOS ONE. After careful consideration, we feel that it has merit but does not fully meet PLOS ONE’s publication criteria as it currently stands. Therefore, we invite you to submit a revised version of the manuscript that addresses the points raised during the review process.

We look forward to receiving your revised manuscript.

Kind regards,

Shanjida Chowdhury

Academic Editor

PLOS ONE

Journal Requirements:

Reviewers' comments:

Reviewer's Responses to Questions

**Comments to the Author**

1. Is the manuscript technically sound, and do the data support the conclusions?

Reviewer #1: No

Reviewer #2: Yes

Reviewer #3: Yes

2. Has the statistical analysis been performed appropriately and rigorously? 

Reviewer #1: No

Reviewer #2: Yes

Reviewer #3: Yes

3. Have the authors made all data underlying the findings in their manuscript fully available?

Reviewer #1: Yes

Reviewer #2: Yes

Reviewer #3: Yes

4. Is the manuscript presented in an intelligible fashion and written in standard English?

Reviewer #1: No

Reviewer #2: Yes

Reviewer #3: Yes

5. Review Comments to the Author

Reviewer #1: The study is severely underdeveloped.

1. There is no research question or policy relevance.

2. Literature review pinpoints no research gap.

3. The empirical model is flawed. It suffers from endogeneity bias (reverse causality and omitted variable).

4. Results are not discussed. They are only reported.

5. There are no implications drawn from the study.

6. The language is below publication standard.

Reviewer #2: 1) The manuscript is well written. The topic is very interesting. However, authors need to revise the abstract and make it more clear and crisp.

2) Add value addition in the introduction, you may read the suggested papers and get the idea how to write down the value addition.

3) Please add these papers in literature review or introduction to make it more rigorous and updated:

1. https://link.springer.com/article/10.1007/s11356-022-21401-9

2.https://link.springer.com/article/10.1007/s11356-022-21339-y

3.https://link.springer.com/article/10.1007/s11356-022-22587-8

4.https://www.sciencedirect.com/science/article/abs/pii/S0301420722003579

5.https://link.springer.com/article/10.1007/s11356-022-23355-4

4) Review the English language and correct grammar and spelling errors.

Reviewer #3: I thoroughly read the research paper entitled “Does Green Innovation Moderate between FDI and Environmental Sustainability? Empirical Evidence from South Asia”. The topic is interesting and will contributes to Environmental Economic literature. Before to publish the paper, revise the paper according to my comments and suggestions.

1. Please clearly elaborate the novelty of your research paper in the introduction section.

2. Please briefly clarify your methodology in the introduction section

3. Please briefly discuss your results at the end of introduction section. So that readers be convenient while read your research paper.

4. The objectives should not be in bulletin. Its should be in paragraph. Its not thesis.

5. The very current literature is missing on Environmental sustainability and FDI in south Asia and global. Please update the literature. For example,

https://doi.org/10.1016/j.apenergy.2023.120836,

https://doi.org/10.1016/j.renene.2023.02.035

https://doi.org/10.1016/j.ribaf.2019.101129

https://doi.org/10.1108/IJOEM-03-2022-0395

10.1061/(ASCE)IS.1943-555X.0000724

DOI 10.1108/JES-03-2020-0123

Good luck!

6. PLOS authors have the option to publish the peer review history of their article (what does this mean?). If published, this will include your full peer review and any attached files.

Reviewer #1: No

Reviewer #2: No

Reviewer #3: **Yes: **Faheem Ur Rehman

While revising your submission, please upload your figure files to the Preflight Analysis and Conversion Engine (PACE) digital diagnostic tool, https://pacev2.apexcovantage.com/. PACE helps ensure that figures meet PLOS requirements. To use PACE, you must first register as a user. Registration is free. Then, login and navigate to the UPLOAD tab, where you will find detailed instructions on how to use the tool. If you encounter any issues or have any questions when using PACE, please email PLOS at figures@plos.org. Please note that Supporting Information files do not need this steps.

<quillbot-extension-portal></quillbot-extension-portal>

---

## [Author Response · Author response to Decision Letter 0]

30 May 2023

Review and Response Sheet

We are very grateful to respected editor and reviewers for their time and efforts to provide the valuable insights for the improvement of this paper. We have revised the manuscript under the suggestions of respected reviewers. The changes are highlighted with yellow color in each part of the manuscript the response of each comment is given below.

Reviewer #1

Comments Author Response

There is no research question or policy relevance. The author has added the research questions. 

The literature review pinpoints no research gap. The study highlighted the research gap

The empirical model is flawed. It suffers from endogeneity bias (reverse causality and omitted variable). Authors has applied the causality test and the for the endogeneity problem authors have employed FMOLS and DOLS estimation.

Results are not discussed. They are only reported.

 Results are discussed in detail.

There are no implications drawn from the study. The author draws Policy implications from the study.

The language is below publication standard. The paper is now proofread by a native speaker.

Reviewer 2

Comments Author Response

The manuscript is well written. The topic is very interesting. However, authors need to revise the abstract and make it clearer and crisper. Thank you for your comment. Author has revised the abstract make it clearer and more understandable. 

Add value addition in the introduction, you may read the suggested papers and get the idea how to write down the value addition.

 Author added the value addition in the introduction and revised the whole intro.

Please add these papers in literature review or introduction to make it more rigorous and updated:

1. https://link.springer.com/article/10.1007/s11356-022-21401-9

2.https://link.springer.com/article/10.1007/s11356-022-21339-y

3.https://link.springer.com/article/10.1007/s11356-022-22587-8

4.https://www.sciencedirect.com/science/article/abs/pii/S

0301420722003579

5.https://link.springer.com/article/10.1007/s11356-022-23355-4

 Thank you for suggesting the valuable papers which increased the empirical evidence and justification. Author has cited the all relevant papers. 

Review the English language and correct grammar and spelling errors.

 The paper is now proofread by native speaker. 

Reviewer #3

Comments Author Response

Please clearly elaborate the novelty of your research paper in the introduction section. Thank you for pointing out the important point. The author has highlighted the contribution of the study.

Please briefly clarify your methodology in the introduction section

 The author has clarified the methodology in the introduction section now it is clearer and more understandable. 

Please briefly discuss your results at the end of introduction section. So that readers be convenient while read your research paper. The author has added the results at the end of the introduction section. 

The objectives should not be in bulletin. Its should be in paragraph. Its not thesis.

 The author has rewritten the objectives according to suggestions. 

The very current literature is missing on Environmental sustainability and FDI in south Asia and global. Please update the literature. For example,

https://doi.org/10.1016/j.apenergy.2023.120836,

https://doi.org/10.1016/j.renene.2023.02.035

https://doi.org/10.1016/j.ribaf.2019.101129

https://doi.org/10.1108/IJOEM-03-2022-0395

10.1061/(ASCE)IS.1943-555X.0000724

DOI 10.1108/JES-03-2020-0123

 Thank you for suggesting the valuable recent papers which increased the empirical evidence and justification. The author has cited all relevant papers.

Thank you,

Sincerely

---

## [Decision Letter · Decision Letter 1]

28 Jun 2023

PONE-D-23-05630R1

Does Green Innovation Moderate between FDI and Environmental Sustainability? Empirical Evidence from South Asia

PLOS ONE

Dear Dr. Brohi,

We’re pleased to inform you that your manuscript has been judged scientifically suitable for publication and will be formally accepted for publication once it meets all outstanding technical requirements.

Kind regards,

Shanjida Chowdhury

Academic Editor

PLOS ONE

- - - - -

<quillbot-extension-portal></quillbot-extension-portal><quillbot-extension-portal></quillbot-extension-portal>

---

## [Editor Report · Acceptance letter]

5 Jul 2023

PONE-D-23-05630R1 

Does Green Innovation Moderate between FDI and Environmental Sustainability? Empirical Evidence from South Asia 

Dear Dr. Brohi:

I'm pleased to inform you that your manuscript has been deemed suitable for publication in PLOS ONE. Congratulations! Your manuscript is now with our production department. 

Kind regards, 

on behalf of

Dr. Shanjida Chowdhury 

Academic Editor

PLOS ONE